



# Including informal housing in slope stability analysis – an application to a data-scarce location in the humid tropics

Elisa Bozzolan[1,2], Elizabeth Holcombe[1,2], Francesca Pianosi[1,2], Thorsten Wagener[1,2]

[1]Department of Civil Engineering, University of Bristol, Bristol, BS8 1TR, UK
[2]Cabot Institute, University of Bristol, Bristol, BS8 1TR, UK

Correspondence to: Elisa Bozzolan (elisa.bozzolan@bristol.ac.uk)

**Abstract**

Empirical evidence from the humid tropics shows that informal housing can increase the occurrence of rainfall-triggered
landslides. However, informal housing is rarely accounted for in landslide hazard assessments at community or larger scales.
We include informal housing influences (vegetation removal, slope cutting, house loading and point water sources) in a slope
stability analysis. We extend the mechanistic model CHASM (Combined Hydrology and Stability Model) to include leaking
pipes, septic tanks, and roof gutters. We test CHASM+ in a region of the humid tropics, using a stochastic framework to
account for uncertainties related to model parameters and drivers (incl. climate change). We find slope cutting to be the most
detrimental construction activity for slope stability. When informal housing is present, more failures (+85%) are observed in
slopes with low landslide susceptibility and for high intensity, short duration precipitations. As a result, the rainfall threshold
for triggering landslides is lower when compared to non-urbanised slopes, and comparable to those found empirically for
similar urbanised regions. Finally, low cost-effective 'low regrets' mitigation actions are suggested to tackle the main landslide
drivers identified in the study area. The proposed methodology and rainfall threshold calculation are suitable for data scarce
contexts, i.e. when not much field measurements or landslide inventories are available.

## 1 Introduction

Global and regional landslide records reveal an increase in rainfall- and human-triggered landslides during the last century,
mainly in economically developing countries with rapid population growth and urbanisation (Kirschbaum, et al., 2015; Froude
and Petley, 2018). This increase might be partly due to continuing improvements in landslide recording, but it also indicates
the growing impact of climate and urban pressure on landslide occurrence (Larsen, 2008). Understanding the mutual
interactions between the natural and urban environment becomes particularly relevant in the humid tropics where high intensity
and duration rainfall events are the main landslide triggers and urban expansion is poorly regulated (Lumb, 1975; UN-Habitat,
2015). The natural landslide susceptibility of these regions coupled with the lack of urban planning and regulations can increase
risk, not only in terms of vulnerability and exposure but also in terms of hazard.

Evidence from low income urban settlements in the humid tropics reveals a link between poorly regulated urban construction
activities and increasing landslide hazard. Potential anthropogenic landslide drivers include slope cutting and filling for house
and road construction (Sidle and Ziegler, 2012; Smyth and Royle, 2000), slope degradation with clearance of forested areas
(Gerrard and Gardner, 2006; Vanacker et al., 2003), and inadequate drainage networks, unplanned redirection of storm runoff
and poorly maintained septic systems (Diaz, 1992; Anderson et al. 2008). In this paper, we use the term 'informal housing' to
refer to the combination of these urban modifications which influence slope stability by altering its geometry, hydrology and
material strength (Figure 1).

However, informal housing is usually neglected or not quantified in landslide hazard assessment at community and larger
scales. There are two main reasons for this: lack of reporting and the highly localised scale and heterogeneous nature of human
landslide drivers. A landslide is defined as triggered by human activities when there is a direct (and easily recognisable)





connection with the failure process (e.g. during mining activities). Landslides of this type are small and often not recorded (Petley, 2012).When considering rainfall triggered-landslides, human landslide drivers are often either not considered or not distinguished from the natural drivers (SafeLand, 2011). Urban construction activities are localised and even if they contribute to land instability, they remain difficult to observe either in situ (e.g. leaking pipes) or via satellite images. For these reasons, there are numerous site specific analysis that investigate the influence of urban construction activities for individual slopes

with known soil and rainfall trigger characteristics (e.g. Preuth et al., 2010; Zhang et al., 2012), but there are few studies that explore the influence of informal housing more widely, for different combinations of human landslide drivers, soils, slope geometry and rainfall triggers. This limits the transferability of the findings from slope to larger scales where less detailed data is available.

Empirical-statistical and heuristic methods have been used in regional studies to link informal housing to the spatial and

temporal occurrence of landslides. For example, precipitation and landslide records have been analysed in relation to lithology and land use change (Alewell and Meusburger, 2008; Gerrard and Gardner, 2006), or in relation to soil type, and type of settlement (Smyth and Royle, 2000). Here, most of the recorded landslides were found to be associated with poorly regulated construction techniques, water management and land degradation. Rainfall thresholds for triggering landslides were observed to depend on impervious surfaces (Diaz, 1992). However, these analyses did not enable the differentiation of the relative role

of natural and human landslide drivers precluding the translation of the results into actions at slope/engineering scale (Anderson et al., 2013; Maes et al., 2017).

Mechanistic slope hydrology and stability models can be used to represent the landslide drivers for historical, current and potential future climate conditions (e.g. Ciabatta et al., 2016; Almeida et al., 2017). If these models included the effect of informal housing, the analysis of different combinations of slope, urban and climate properties could lead to assess the relative

role of natural and urban properties on triggering landslides and to identify the conditions at which urban construction activities become most detrimental. This could be a useful information for engineers to prioritise slopes that are currently at risk, to identify those at higher risk to be impacted in the future, and to deduce appropriate hazard mitigation or preparedness actions. The inclusion of informal housing in slope stability analysis could also lead to considerations about the reliability of rainfall thresholds for triggering landslides within highly urbanised communities, since they might be underestimating the level of the

hazard (Mendes et al., 2018).

However, the use of data intensive mechanistic models can be challenging in data scarce locations, such as in low income urban settlements. The more complex the model, the more data required to set its parameters and model forcing, and the more uncertainties might be introduced into the analysis. Sources of uncertainties can relate to slope and soil properties, urban features as well as to a limited understanding of physical processes or future scenarios (epistemic uncertainties) (see Beven et

al., 2018a, for a review of this issue). Many researchers have assessed the impact of uncertainties related to slope properties (e.g. Cho, 2007) and future climate (e.g. Ciabatta et al., 2016) on slope stability at different scales. However, to the best of our knowledge, there are no analyses that consider both sources of uncertainties when modelling informal housing in landslide hazard assessment. Urban construction activities are either considered separately (e.g. slope cutting or pipes leaking) (e.g. El-Ramly et al., 2006) or the slope properties are varied using discrete conservative values under fixed rainstorm conditions

(Anderson et al., 2008; Holcombe et al., 2016). This separation might overlook significant changes of the slope's behaviour for combinations of urban constructions activities and/or slope/soil/rainfall properties that have not been considered but are still likely to occur.

Almeida et al., (2017) has demonstrated how mechanistic landslide models can consider both uncertainties due to poorly defined slope properties and to potential future climate changes. The mechanistic model CHASM (Combined Hydrology and

Stability Model) was used in a Monte Carlo framework and applied in Saint Lucia, in the Eastern Caribbean, where data support is limited but landslide hazard is particularly high. The probability distributions of slope and soil properties were extrapolated from available data and literature, while the rainfall properties were varied widely and uniformly, also considering





rainfall intensity-duration combinations which might be observed in the future. Statistical and data mining algorithms were then used to quantify the relative role of the input factors (and thus their uncertainties) on the stability of the simulated slopes

as well as to identify critical thresholds in slope properties and rainfall drivers likely to lead to slope failure.

In this study we extend the work of Almeida et al. (2017) by including informal housing into such a slope stability analysis. We consider the same location of the humid tropics and the same core model, CHASM, but with new functions to represent the mechanistic influences of informal housing. CHASM is a two-dimensional model which has a relative low data requirement for a mechanistic model even with the inclusion of the new informal housing functions. In addition to the original ability to

represent the mechanical and hydrological effects of vegetation and the effects of slope cutting and loading, we have added the effects of point water sources resulting from leaking septic tanks, water supply pipes, and houses without roof gutters. By varying both the natural and urban factors, we aim to identify under which slope and climate conditions landslide hazard is significantly increased by the presence of informal housing and how this information can be used for deducing landslide mitigation measures Thus, for our humid tropical case study scenario we aim to address the following questions:

1. How can we identify which informal urban housing characteristics are most detrimental to slope stability?

2. How is the rainfall threshold for triggering landslides modified when informal housing is considered?

3. Which landslide mitigation strategies and practices can be deduced from the analysis for current and potential future scenarios of urbanisation and rainfall?

The proposed methodology is suitable for data scarce contexts, i.e. when not much field measurements or landslide inventories

are available. If applied in countries with similar natural/climate/urban characteristics (so with similar input space variability) we might expect similar slope stability responses and thresholds. Conversely, a change in (part of) the input data (or their probability distributions) to reflect a different urban landslide context could potentially produce quite different outputs (Wagener and Pianosi 2019).

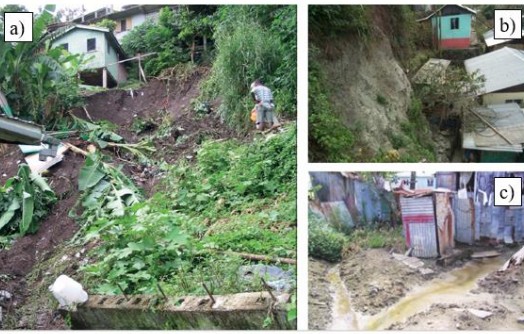

**Figure 1: Examples of informal housing affecting land stability. The image (a) and (b) show examples of unsupported cut slopes, respectively in Saint Lucia (Caribbean) and DumsiPakha (Kalimpong, India). The image (c) shows the effect of lack of water management in an informal community of Saint Lucia (Caribbean). From the blog Save the Hills (http://savethehills.blogspot.com/) and the community based project Mossaic (Anderson and Holcombe, 2013)**

## 2. Method

We want to analyse the relative role of informal housing on slope stability under different natural and climate conditions. The methodology we introduce here entails the following steps:

- Choose a model that represents the main instability mechanisms of the case study area. We are interested in representing the rainfall-triggered landslides and the informal housing of Saint Lucia (Caribbean). We therefore use the mechanistic model CHASM which represents both the hydrology-stability routing, but also vegetation, slope

cutting, and various forms of water management.



- Define the inputs factors necessary to run the model and their variability space. In our case study, the input factors are the parameters defining the slope soil, geometry, urban characteristics, as well as rainfall forcing data. Each input factor is assumed to be a random variable and its range of variability is determined by a probability distribution. The probability distributions can be defined based on the physical meaning of the input factors, available data and/or existing literature. We use information gathered both from fieldwork in Saint Lucia and also from literature.

- Create synthetic combinations of input factors by stochastically sampling from their probability distributions and run CHASM to generate an equivalent number of model outputs. We select the minimum Factor of Safety (FoS) and the slip surface where the minimum FoS is calculated as summary output variables to analyse. We repeat the stochastic sampling with and without including the urban properties among the input factors, in order to facilitate considerations about the role of informal housing on land stability.

- Identify the input factors that most influence slope stability using global sensitivity analysis (Wagener and Pianosi, 2019). In particular, we use a regional sensitivity analysis approach (RSA, Hornberger and Spear, 1981) to identify which input factors are most influential in leading to slope failure.

- Identify parameters' thresholds beyond which the slopes become unstable. The threshold of an input factor over/below which failure is predicted might depend on the value of the other input factors (e.g. slopes with higher slope angles require higher soil strength to maintain stability). Machine learning is a set of methods that computers use to understand trends from data, also considering their mutual interactions. We use CART (Classification and Regression Trees) to develop a set of decision rules that predict for which combination of soil, geometry, urbanisation and rainfall input values a particular slope is more likely to fail.

In the following paragraphs we are going to describe in detail the tools and the data used to implement our analysis on the island of Saint Lucia.

### 2.1 The study site: Saint Lucia, Eastern Caribbean

Saint Lucia is an Eastern Caribbean island in which informal housing has led to an increase of landslide risk (WB/GFDRR 2012, p. 226-235). Much of the housing is built using unregulated engineering practices in a region that is already susceptible to landslides due to its mountainous topography and predominantly deep volcanic residual soils. The main landslide trigger is rainfall and shallow rotational landslides dominate on both steep and shallow slopes (Van Westen, 2016; Anderson and Holcombe, 2013). The typical slope cross section is composed of soil strata of weathered residual soils overlying decomposed rock and volcanic bedrock. Various sources of information on the slope, soil, rainfall and urban properties of this region are available from previous studies by government engineers and planners, the local water company and consultants (e.g. CHARIM, 2015; Mott MacDonald, 2013; Klohn-Crippen, 1995), and from community based projects for the improvement of slope stability with surface water drainage works (Anderson and Holcombe, 2013). In this project, estimates of soil strength properties are based on direct shear tests of local soils (Anderson and Kemp, 1985; DIWI, 2002; Holcombe, 2006), and secondary data sources on similar volcanic tropical residual soils such as those in Hong Kong (Anderson, 1982; Anderson and Howes, 1985). Information about soil type, soil depth, type of house construction, cut slope angles and the management of surface runoff and waste water on slopes was based on community-based mapping and elicitation of local expert knowledge undertaken by Anderson and Holcombe (2013) who co-developed these datasets with residents, government, and local experts.

### 2.2 CHASM: a mechanistic model for rainfall-triggered landslides

CHASM (Combined Hydrology and Stability Model) is a 2-D mechanistic model which analyses dynamic slope hydrology and its effect on slope stability over time. A full description of the model can be found in Anderson and Lloyd (1991) and Wilkinson et al. (2002a,b). Here we briefly describe its hydrology and stability components, whereas the representation of the urban properties is detailed in Paragraph 2.3. In CHASM the slope cross section is represented with a regular mesh of columns


and cells. Hydrological and geotechnical parameters are specified per cell, while the initial hydrological conditions define the position of the water table, and the matric suction of the top cell of each column. The dynamic forcing for CHASM is rainfall specified in terms of intensity and duration. For each computational time step (usually 10−60s), a forward explicit finite

difference method is used to solve the Richard's (1-D, vertical flow) and Darcy's (2-D flow) equations which regulate respectively the unsaturated and saturated groundwater flow. At the end of each simulation hour the resulting soil pore water pressures (positive and negative) are used as input for the slope stability analysis which implements a Bishop's simplified circular limit equilibrium method (Bishop, 1955) and uses the coordinates of the slope surface. An automated search algorithm identifies the location of the slip surface with the minimum factor of safety, FoS, which is given as output at the end of each

simulated hour. In a validation exercise in Hong Kong, CHASM shown an accuracy of 72.5% (Anderson, 1990) which is comparable to the performances of other models used for the stability analysis (e.g. Formetta et al., 2014). CHASM has been employed in Malaysia, Indonesia, Eastern Caribbean, and New Zealand, to propose landslide mitigation measures, as well as to identify land instability drivers along roads and in urban and rural areas (Brooks et al., 2004; Lloyd et al., 2001). Almeida et al. (2017) used CHASM stochastically in a Monte Carlo framework to account for uncertainties in both slope properties and

future climate scenarios.

### 2.3 A new functionality in CHASM: urban point water sources

The new CHASM+ can now not only represent slope cutting, additional (house and tank) load, and vegetation removal, but also the presence or absence of roof gutters on houses and localised water leakages from buried septic tanks and superficial water supply pipe networks. Slope cuttings are represented by a corresponding change in slope geometry; additional loads are

simulated by appropriately increasing the unit weight of the soil underneath the loading object (i.e. house and tanks); vegetation, which is removed during the urbanisation process, is represented through rainfall interception, evapotranspiration, root water uptake, vegetation surcharge, and increased permeability and soil cohesion due to the root network (see Wilkinson et al., 2002b). Pipes above ground and buried tanks can be added to the slope, with specified dimensions and leakage rates. Pipe leakage is accounted for as additional surface water which infiltrates into the slope according to the infiltration capacity

of the soil. If water exceeds the infiltration capacity of the soil, it is stored as ponding water. If the ponding water exceeds the maximum water detention capacity (set at 10 mm), the water excess is removed (no runoff considered). Leakage from tanks is added to the water moisture content in the soil cells underneath the tank. Where houses are present, rainfall is intercepted by the roof. If roof gutters are not included, the intercepted rainwater is discharged onto the slope cells adjacent to the house, in accordance to the roof type (double or single pitch). More details on the new functionality and its benchmarking against another

model are given in the supplementary document that accompanies this paper (S1.1. and S1.2).

### 2.4 Definition of the input factors and their probability distributions

We use thirty input factors to characterise our case study area in CHASM+. These factors fall into the following categories: slope profile geometry, soil geotechnical and hydrological properties, urban characteristics, initial hydrological conditions and rainfall properties. Table 1 reports the full list of these input factors and the probability distributions that define their range of

variability, while Fig. 3 shows an example of slope derived from a combination of input factors.

The slope geometric properties consists of the natural slope angle and height, and the material thickness. Slope angles vary between 20 and 45 degrees to represent typical scenarios of informal housing on moderate and steep slopes. 45° is considered the highest slope angle on which a settlement can be located without some form of engineered slope stabilisation measures. The cross-sectional profile is discretised into three parallel layers of materials to represent the typical weathering profile of

volcanic parent material, with a layer of residual soil at the surface (layer 1), underlain by a layer of weathered material (layer 2), and then unweathered bedrock (layer 3). Ranges of material thickness and geotechnical properties are derived from previous field work and lab tests, as described in section 2.1.



The height of the water table is defined as an initial hydrological condition. This water table height is varied between 0% and 90% of the slope height (H in Fig. 3), to account for the uncertainties relating to the initial soil moisture conditions, including

the potential effect of antecedent rainfall events.

The model is forced with rainfall events which are specified in terms of their duration (in hours) and hourly intensity. Rainfall value ranges are based on the intensity-duration-frequency (IDF) relationships derived from a Gumbel analysis of 40-years of daily rainfall data from weather stations across the island by Klohn-Crippen (1995). From these IDFs we define ranges of rainfall intensity from 0 to 200 mm h$^{-1}$, and duration from 0 to 72 h which are sampled independently from uniform

distributions. In this way we take into account our poor knowledge about future climate by including any possible future design storm in the sampling (Almeida et al., 2017). Prior to the initiation of the rainfall event we include 168 simulation hours (7 days) with rainfall intensity equal to zero. This ensures a redistribution of water moisture in the unsaturated zone of the slope and allows hydrological equilibrium with steady state seepage to be established. A further 168 hours of zero rainfall simulation are added after the storm in order to consider the ground water response after the rainfall event.

Informal housing is represented by four urban properties: slope cutting, absence of roof gutters, vegetation removal, and leaking pipes and tanks. While the angle of the cut slope is varied according to its probability distribution, the vegetation, roof gutters and water leakage are defined as present (option 1 = yes) or absent (option 0 = no) (Fig.2). The cut slope angle is varied between 39 and 89 degrees with a maximum cut slope height equal to 4m. We represent the maximum number of cut slopes that can accommodate a house that is 4m wide (+1m of surrounding space) on a slope that is 70m long. We therefore obtain

either five or six cut slopes and a corresponding number of houses on each slope depending on the angle of the cut slope. The house width and house load (8 kN m$^{-2}$) are not varied, and correspond to the size and load of informal houses constructed with shallow concrete strip or block foundations, wooden walls and sheet-metal roofing that are typically observed in Saint Lucia (Holcombe et al., 2016). When vegetation is present on the original non-urbanised slope, it is removed on the surface of the cuts for the urban scenario. The vegetation properties used represent a tropical forest cover, a sensitive choice for this study

site (see Holcombe et al. 2016 and online supplementary material, Table S5). These properties are kept fixed throughout the sampling, therefore the effect of different types of vegetation on slope stability is not analysed. Both the tank and the pipe leakage rate are assumed to be half of 4.2e$^{-6}$ m$^3$ s$^{-1}$, which corresponds to the estimated leakage of 15% of the total water supply for low income households in Saint Lucia (Anderson and Holcombe 2013). When present, the leak is maintained constant during the simulation time.

The input factors that define the discretisation of the model, such as the cell size of 1m x 1m and the computational time step of 60 s (both used by CHASM's dynamic hydrology functions), and the slip search grid location and dimensions, are not varied. These values are chosen because they typically ensure numerical stability relating to the mass balance of the moisture in the domain and thus a minimum number of failed model's runs. A smaller cell-size would enable a more detailed representation of the slope hydrology, but it would require smaller time step to preserve the moisture content mass balance

and numerical stability. Smaller time steps would result in significantly longer simulation time. The resolution chosen is therefore a trade-off between acceptable accuracy and calculation time. The influence of the variation of these two discretisation parameters on slope stability is not explored.

### 2.5 Creation of synthetic combinations of input factors and model simulation

We use Latin Hypercube sampling to generate 10 000 different combinations of the 30 independently varying input factors

shown in Table 1. Figure 3 illustrates how each slope is defined based on these input factors combination. Due to the randomness of the process, checks are undertaken to ensure that realistic combinations of factors are generated; if not, they are discarded (around 70% of the times) and replaced by another randomly generated, feasible combination These "feasibility" checks are reported at the footnote of Table 1(letters a, b, c, d and f). The combinations of input factors are then run in CHASM+ using the high performance computer BlueCrystal Phase 3 at the University of Bristol. The outputs considered for each




simulation are the minimum Factor of Safety (FoS) and the slip surface where the minimum FoS is calculated. We divide the

simulations in failed or stable according to the value of the minimum FoS (failed if FoS less than 1). We exclude the simulations

that failed before the start of the rainfall event, which represent inherently unstable slopes (for example steep slopes with deep

soil thickness). We repeat the same procedure with and without including the urban properties. We therefore obtain two sets

of model's outputs: 10 000 representing urbanised slopes, and 10 000 representing non-urbanised slopes.

**Table 1: Input factors of CHASM and their probability distributions**

| Parameter | Symbol/Unit | | Range values | | |
|---|---|---|---|---|---|
| **Slope geometric properties:** | | | **Layer 1 \*** | **Layer 2 \*** | **Layer 3 \*** |
| Slope angle | δ [degrees] | U (20,45) | | | |
| Thickness of layer | H [m] | | U (1,6) | U (1,6) | |
| **Soil properties:** | | | | | |
| Effective cohesion [a] | c [kPa] | | Ln (2.3688, 0.5698) | Ln (3.4121, 0.5774) | 80 |
| Effective friction angle [b] | φ [degrees] | | Ln (3.2937, 0.2092) | Ln (3.1559, 0.3251) | 60 |
| Dry unit weight [c] | γd [kN m$^{-3}$] | | U (16,18) | U (18, 20) | 23 |
| Saturated moisture content [d] | VG θsat [m$^3$ m$^{-3}$] | | N (0.413, 0.074) | N (0.413, 0.074) | N (0.413, 0.074) |
| Residual moisture content [d] | VG θres [m$^3$ m$^{-3}$] | | Ln (-1.974, 0.376) | Ln (-1.974, 0.376) | Ln (-1.974, 0.376) |
| VG alpha parameter [d] | VG α [m$^{-1}$] | | Ln (1.264, 1.076) | Ln (1.264, 1.076) | Ln (1.264, 1.076) |
| VG n parameters [d] | VG n | | Ln (0.364, 0.358) | Ln (0.364, 0.358) | Ln (0.364, 0.358) |
| Saturated Hydraulic Conductivity | Ksat [m s$^{-1}$] | | Ln (-11.055, 0.373) | Ln (-13.357, 0.373) | 1xe-8 |
| **Initial hydrological condition** | | | | | |
| Water table height [e] | DWT [%] | U (0,90) | | | |
| **Rainfall properties** | | | | | |
| Rain intensity | I [m s$^{-1}$] | U (0 0.2) | | | |
| Rain duration | D [h] | Ud (1 72) | | | |
| **Urban properties:** | | | | | |
| Cut slope angle [f] | β [degrees] | N (65.18, 12.61) | | | |
| Roof gutters [g] | - | Ud (0 1) | | | |
| Vegetation [h] | - | Ud (0 1) | | | |
| Septic tank and Pipe leak [i] | Qt/p [m$^3$ s$^{-1}$] | Ud (0 1) | | | |

U = Uniform distribution; Ud = Discrete uniform; N = Normal distribution; Ln = Log-normal distribution.
\*Layer 1: Residual Soil, Weathering Grade V-VI; Layer2: Weathered material Grade III–IV; Layer3: bedrock Grade I–II; Weathering grades defined according to GEO (1988).

[a] Effective cohesion > 0. Effective cohesion c (layer 3) > c (layer 2)> c (layer 3).

[b] Effective friction angle > 0. Effective friction angle φ (layer 3) > φ (layer 2)> φ (layer 3). φ < 90 degrees

[c] γs =γd +2, where γs is the saturated unit weight. γd (layer 3) > γd (layer 2) > γd (layer 1)

[d] Values from Hodnett and Tomasella (2002) for Sandy Clay Loam material. We impose n > 1; θsat > θres; θres > 0.
VG: Van Genuchten parameters for defining suction moisture characteristics curve.

[e] Water table height is defined as a percentage of slope height measured to the toe of the slope.

[f] Slope of the cut forced to be between 39 and 89 degrees, and it is always greater than natural slope angle

[g] 0 stands for house without rain gutters; 1 stands for house with rain gutters. Roof type = double pitch

[h] Vegetation presence: 0 no vegetation; 1 insert vegetation in the spare spaces.

[i] The leak of the septic tank is equal to the leak of the pipe. When 0 is selected there is no leak, whilst with 1 there are both. The leak rate is constant and equal to 4.2e-6 m$^3$ s$^{-1}$

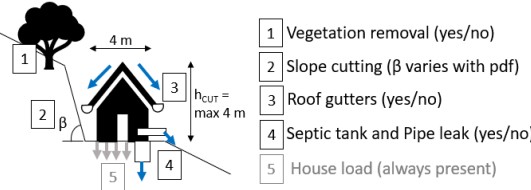

1 Vegetation removal (yes/no)
2 Slope cutting (β varies with pdf)
3 Roof gutters (yes/no)
4 Septic tank and Pipe leak (yes/no)
5 House load (always present)

**Figure 2: Urban properties of informal housing included in the slope stability analysis. Each house corresponds to a cut on the slope. Cut slope angle varies according to its probability distribution, defined in Table 1. Vegetation, roof gutters, leaking tanks/pipes are**
**stochastically inserted or not. The house on the cut slope is always present and its load is not varied. The height of the cut slope varies relatively to the cut slope angle, but it is forced to be maximum 4 metres**




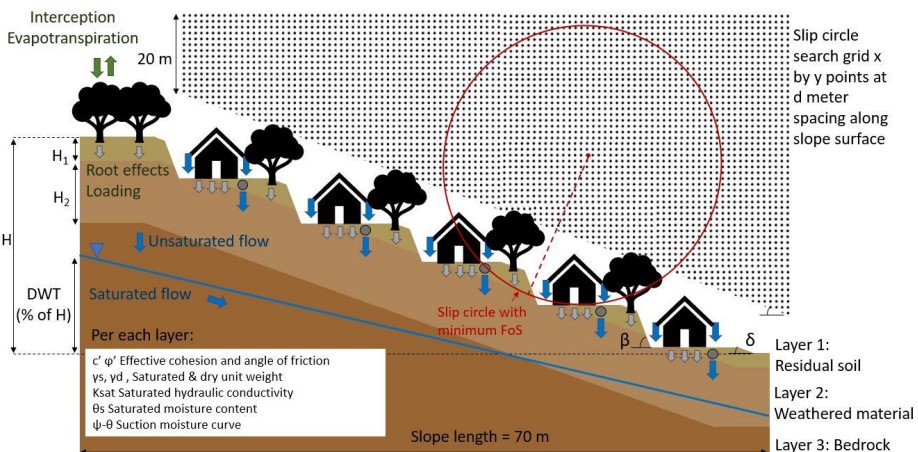

**Figure 3: Example of slope generated by stochastically sampling from the ranges of input factors specified in Table 1. H is the slope height resulting from the fixed slope length and varying slope angles. The dimensions of the slip circle search grid are fixed, with**
**initial height of 20 m, and width equal to the slope length. The grid extends downslope parallel to the slope as shown.**

### 2.6 Regional sensitivity analysis (RSA) and Classification And Regression Trees (CART)

Global sensitivity analysis is a set of statistical techniques that evaluate how the variations of a model's outputs can be attributed to the variations of the model's input factors. In this case we want to identify which input factors - among geometry, soil, hydrology, rainfall and urban properties - have the strongest impact on slope stability. Since in our case the model output
is binary, as simulated slopes are categorised as failed (if FoS<1) or stable (FoS>1), we use the Regional Sensitivity Analysis (RSA) approach (Hornberger and Spear, 1981) which is particularly suitable when dealing with categorical outputs. In the RSA approach, the cumulative marginal distribution of each input factor is computed for each output category, i.e. the stable slopes and the failed ones. If the distributions significantly separate out, it is taken as evidence that the model output (slope stability) is significantly affected by variations of the considered input factor. The level of separation between the cumulative
distributions can be formally measured with the Kolmorov-Smirnov (KS) statistic and used as sensitivity index. The confidence intervals of the sensitivity indexes can be estimated via bootstrap technique. The bootstrap randomly draws N samples (with replacement) from the available data, to compute N KS statistics for each input factor. The magnitude of fluctuations of the KS statistic from one sample to another represents the level of confidence in the estimation of the sensitivity indexes. For this application, we use the SAFE toolbox (Pianosi et al. 2015) to perform RSA and to calculate the sensitivity indices and their
confidence intervals by bootstrapping technique.

Classification And Regression Tree (CART) analysis is a supervised machine learning method which we use to predict critical thresholds in input factors over/below which a particular slope is more likely to fail (Breiman et al., 1984). In this analysis, the predictor model takes the form of a binary tree. Starting from the whole set of simulations, CART finds the best possible input factor (e.g. slope angle rather than rainfall intensity) and the best possible value of that input factor (e.g. slope angle greater or
less than 30°) that divide the simulations into stable and failed simulations. This process is recursively repeated, creating at every split two branches and two ("child") nodes of the tree. In choosing the best splitter the model seeks to maximise the "purity", i.e. to maximise the number of stable or failed simulations at the two generated nodes. Amongst the different measures of purity available, we use the Gini purity index defined as:

$$1 - \sum_{i=1}^{m} p^2(i) \tag{1}$$

where $m$ is the number of categories for the output (in this case two: stable or failed) and $p(i)$ is the fraction of simulations in
the node belonging to category $i$. The Gini purity index is 0 when all the simulations in the considered node belong to the same



category (a "pure" node, i.e. all stable or failed). The splitting process typically continues until all final leaf nodes show Gini purity indices below a chosen threshold. The final nodes express the prediction for the corresponding branch. While a high number of nodes increases predicting accuracy, it also makes the model more difficult to interpret and generalise to other datasets (i.e. the problem of overfitting). A pruning technique can be applied to avoid this overfitting and to identify an

acceptable trade-off between predictive power and number of nodes. In this analysis, we build a CART using the Matlab Statistics and Machine Learning Toolbox (Matlab R2018a), using the K-fold cross-validation to better estimate its predictive power. In particular, we use 10-fold cross-validation, which randomly divide the original dataset (10 000 simulations) into 10 sub-groups. 9 sub-groups are used to construct 10 CARTs, while the remaining sub-group is used to test the CARTs performance. The average value of the ten misclassification errors so obtained, represents the cross-validation error which can

be used to select suitable pruning levels. To reduce the number of nodes without increasing the misclassification errors, auxiliary variables can be used to combine correlated input factors. Auxiliary variables can simplify the tree's structure (by using fewer combined input factors) and potentially modify the input space in a way that the division of failed and stable simulations is more effective (see rotation of the coordinate systems in Dalal et al., 2013). Three auxiliary variables will be used in this analysis: the ratio of soil thickness and effective soil cohesion of layer 1; the ratio between rainfall intensity and

duration introduced (both introduced by Almeida et al. 2017) and a weighted combination of natural and cut slopes angles. These variables will be described in the results (CART analysis) section and supplementary material (S2).

## 3 Results

In this section we analyse the 10 000x2 outputs generated by CHASM+ for the urbanised and non-urbanised slope scenarios. As previously mentioned, we split the simulations into stable and failed according to the value of the minimum FoS

(respectively greater or less than one). As a first analysis we compare the percentage of failed slopes against stable slopes for each of the urban properties. Figure 4 shows that the presence of cut slopes significantly influences the percentage of predicted slope failures: the steeper the cut slope angle, the higher the percentage of failed slopes. Vegetation removal and roof gutters instead have a negligible role in dividing the two sets. Last, septic tanks and leaking pipes have some effect, generating about 10% more failed slopes when present.

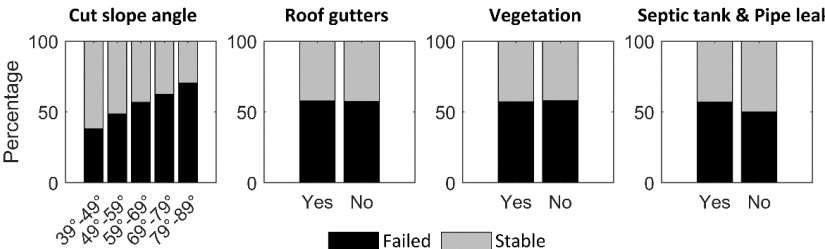


**Figure 4: Percentage of predicted stable and failed slopes per each urban property. An urban property will be influencing slope stability if the percentage of the predicted failures changes with the variation of that urban property.**

### 3.1 Regional Sensitivity Analysis

We then perform RSA on both sets of urbanised and non-urbanised slope simulations, calculating the cumulative marginal

distributions of the failed and stable simulations for each input factor. The maximum distance between the two distributions (KS statistic) is computed and used as a sensitivity index. A high value of the sensitivity index suggests that the variation of that input factor significantly influences slope stability. The results are shown in Fig. 5, for both urbanised and non-urbanised slopes. Figure 5 shows that slope stability is insensitive to many input factors, and highly sensitive to few, namely effective cohesion and thickness of the layer 1 (residual soil), slope angle, and rain intensity and duration. These sensitive input factors



represent the main landslide drivers. The sensitivity indices of the urban properties (in orange) are consistent with the findings of Fig. 4, where only the variation of cut slope angle influences slope stability. When looking at the comparison between urbanised and non-urbanised slopes, it appears that the urban presence decreases the sensitivity indices of all the input factors, except for the effective cohesion of layer 1 and the rainfall intensity.

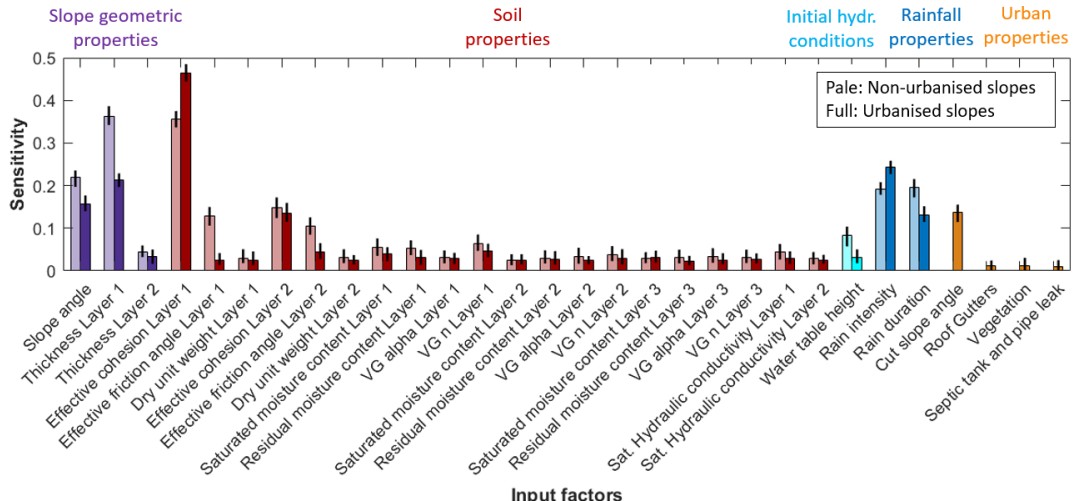


**Figure 5: Sensitivity index for each input factor in the urbanised (full colour) and not urbanised (pale colour) cases. The bars correspond to the mean value of sensitivity for each input factor calculated with bootstrapping, while the black vertical lines at the top of the bars represent the confidence interval (Number of bootstrap resampling N = 100; significance level for the confidence intervals 0.05).**

We further explore the change in sensitivity caused by urbanisation by plotting the percentage of failed slopes for the main landslide drivers (Fig. 6). The figure shows how this percentage varies for the urbanised (black bars and lines) and non-urbanised cases (green bars and lines). In general, urbanised slopes produce more failures than non-urbanised slopes though they both display similar trends: an increased percentage of predicted landslides when we would expect the slope to become more susceptible (e.g. when slope angles are higher) or the trigger more severe (when rainfall intensity and duration are larger).

For example, in Fig. 6b the percentage of failed slopes in the non-urbanised case, linearly increases from ~5% (for soil thickness 1–2m) to ~50% (thickness of 5–6m). In the same figure, urbanised slopes show higher failure rates for all values, though the greatest increase occurs for soil thicknesses less than 4 metres (up to +46% for category 2–3m). This means that the most significant increase in number of landslides occurs for thin soil thicknesses, i.e. on slopes less susceptible to failure when non-urbanised. The same can be said for slope angles less than 25 degrees and rainfall duration less than 10 hours, where

percentages of slope failures passes from less than 15% to more than 40% when urbanisation is introduced (Fig 6 a,c). In the lower plots instead, more urban landslides are observed on slopes that show high percentage of failures also when urbanisation is not present (+43% for low values of soil cohesion, Fig. 6d; +35%, for high rainfall intensities Fig. 6e).The difference in failure rates with variations of input factors also explains the change in sensitivity found in Fig. 5: when urbanised, slope's response varies less (less sensitive) to variations of the input factors in the upper plots (whose sensitivity indices gets smaller),

and more (more sensitive) to variations of the input factors of the lower plots (whose sensitivity indices gets larger).

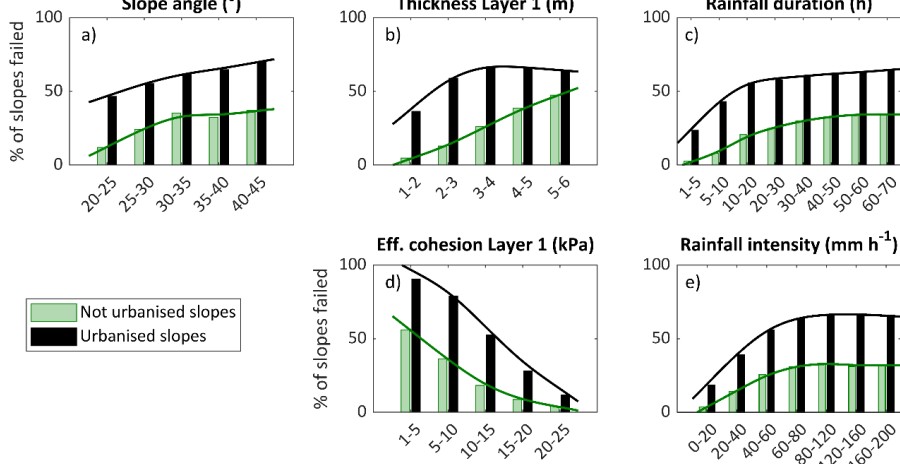

**Figure 6: Percentage of slope failures for urbanised and non-urbanised slopes for different categories of input factors. Throughout, urbanised slopes show higher failure rates than non-urbanised slopes. In the upper plots (a), (b), (c), the distribution of failure rates for urbanised slopes are more uniform for variations of input factors than the non-urbanised case, while in the lower plots (d), (c) it is more pronounced. The upper plots represent the input factors whose sensitivity indexes are smaller when urbanisation is introduced in Fig.5, while the lower plots show the input factors whose sensitivity gets smaller when urbanisation is introduced in Fig.5**

### 3.2 CART analysis

We use the CART analysis to formalise the critical thresholds of input factors above/below which slopes are most likely to be predicted as stable or failed. Figure 7 represents the two trees for the non-urbanised (a) and urbanised case (b). As expected, the best predictor selected in CART are the same input factors previously identified as most influential (Fig.5). The boxes with double colour represent the auxiliary variables that combine correlated input factors: the ratio between effective cohesion and thickness of layer 1 to account their counterbalancing effect on slope stability (i.e. slope with more cohesive soil can be thicker

without experiencing failure); the negative ratio between the logarithm of rainfall intensity and rainfall duration, which represent the slope of the rainfall threshold for triggering landslides; and the weighted average of the natural and the cut slope angles, to account that slope susceptibility can significantly increase for low natural slope angles but high cut slopes angles (see Section 2 of the supplementary document for details about the auxiliary variables and the change in model's performance when they are not considered). Using these few predictors, both trees correctly classify more than 85% of the simulations as

stable or failed (details about the pruning in Section 3 of the supplementary document). Each branch of the tree shows the paths and thresholds of input factors that lead to slopes most likely to fail (black branch), or most likely to not fail (grey branch). At the end of each branch the black/grey bar shows the fraction of failed and stable simulations, while the thickness of the branch is proportional to the number of simulations following that path. For example, in the tree resulting from non-urbanised slopes (left hand side), the thickest grey line shows that more than 50% of simulated slopes resulted stable 91.2%

of the times for ratios of cohesion/thickness of layer 1 greater than 2.5 kPa m[-1]. The thick black branch instead shows that the greatest proportion of simulations predicted as failed occurred for ratios of cohesion/thickness of layer 1 less than 2.5 kPa m[-1], rainfall intensity duration ratios (-log(I)/log(D)) greater than 0.9 m h[-2] and slope angles greater than 25 degrees.

In the trees resulting from non-urbanised slopes (right hand side), the black branch leading to the majority of failures is similar to the non-urbanised tree, but it presents higher splitting thresholds: from the top, the split happens for ratios of

cohesion/thickness of layer 1 less than 4.9 (instead of 2.5) and for rainfall intensity/duration ratio 1.06 (instead of 0.9). The branch then leads to the majority of failures for values of effective cohesion of layer 1 less than 12.6 kPa, regardless of the





natural slope angle. Higher threshold in cohesion/thickness ratios indicates that when urbanisation is present, more failures occur on slopes with higher soil cohesion and/or thinner soil layers than non-urbanised slopes (compatible with Fig. 6b and 6d), while higher rainfall intensity duration ratios suggest that more failures occur for higher rainfall intensity and/or lower

rainfall durations when compared to non-urbanised slopes (as shown in Fig. 6c and 6e). Finally, going back to the top and looking at the grey thick branch of the urbanised tree, it can be noted that a ratio between the effective cohesion and the thickness of layer 1 greater than 4.9 ensured 95% of slope stability only when the weighted slope angle is less than 48 degrees.

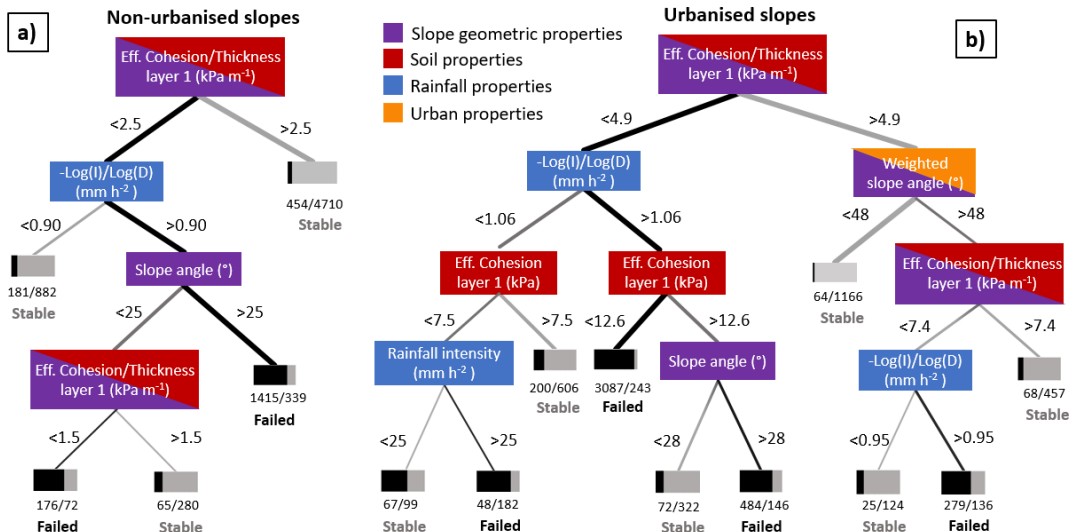

**Figure 7: Classification tree of slope response for non-urbanised slopes (a) and urbanised slopes (b). Black branches represent the paths that lead to simulations predicted as failed, while grey branches lead to simulations predicted as stable. The bar under each leaf shows the proportion of simulations that resulted as failed (black) or stable (grey) for that leaf. The thickness of the branch is proportional to the number of simulations following that path. Note as 14% and 22% of the simulated slopes have been excluded respectively for the non-urbanised and urbanised case, because failed before the start of the precipitation.**

**4 Discussion**

**4.1 Slope cutting is the urban construction activity most detrimental for slope stability**

 In this analysis, slope cutting is the urban construction activity with the strongest effect on slope stability's response (Fig.4 and Fig.5). Figure 6 indicates that when urbanisation is present, more slopes failures are observed, mainly on slopes with relatively low slope angles, low topsoil thickness of layer 1 and low cohesion values (Fig. 6b,d, also reflected by higher

effective cohesion/ thickness ratios in CART in Fig.7b). This is interpreted as caused by cut slopes: when cut slope angles are steep, a higher effective cohesion, and thus a higher soil strength, is required to maintain stability, regardless the natural slope angles; when soil layers intersect the cuts, small soil strength resulting from thin soil layers is not sufficient to insure slope stability. The intersection between soil layer 1 and cut slopes is deduced from Fig.6b: almost 50% more failures are observed for thickness of layer 1 smaller than the slope's height (4 m), i.e. when the layer intersects the cut (as illustrated in Figure 3).

For these slopes, visual inspection reveals that the slip surface is generally located between layer 1 (residual soil, weathering grade V – VI) and layer 2 (weathered material, grade III – IV). This is explained by the different soil strength of the two layers, which constrains the slip surface within the weaker layer 1, and the different hydraulic conductivities. As rainfall infiltrates, the lower hydraulic conductivity of the underneath weathered material leads to a progressive accumulation of water, promoting perched groundwater table. The raised pore water pressure decreases the effective soil strength and consequently the stability

of the soil layer.



Slope cutting is therefore considered, in this analysis, the most detrimental practice for slope stability. This result is consistent with studies carried out in the humid tropics at regional scales, for which slope cutting was identified as one of the major cause of landslides (e.g. Brand et al., 1984; Froude and Petley, 2018; Holcombe et al., 2016). Cuts with slope angles greater than 60° are also considered at particular high risk (e.g. Cheng, 2009), while excess of pore water pressure was shown to be a

dominant process in triggering shallow failures on cut slopes (Anderson, 1983). CHASM therefore successfully captures these physical mechanisms, confirming, despite the uncertainties, the governing role of soil properties and soil thickness in determining slope equilibrium. The other urban construction activities considered, seem to have a less significant role on landslide hazard. Previous studies found that vegetation can be both beneficial and detrimental for slope stability (Wu et al. 1979; Collison and Anderson, 1996). Here we find that its effect is negligible, probably due to its limited presence in urbanised

slopes (trees are left at the crest of each cut slope where they add loading and may actually be detrimental to the local cut slope stability). Also, adding roof gutters does not seem to decrease the number of slopes failed. However, in the scenarios generated here we have only reached a maximum of 30% slope coverage by houses, i.e. about 30% of impervious surface (5–6 households on 70 metres slope), due to our inclusion of cut slopes for every house. Evidence shows that roof guttering effectiveness become evident only when the house coverage is above 50%, and thus a considerable portion of rain does not infiltrate into

the slope (Anderson and Holcombe, 2013). On the other hand, leak from septic tanks and pipes lead to 10% more failures despite the low house coverage. When higher house densities are considered, the lack of water management might become even more significant (Di Martire et al., 2012).

### 4.2 The rainfall threshold for triggering landslides is lower when informal housing is included

We found that when slopes are urbanised, the most significant increase in the percentage of slopes failed occurs for rainstorm
events with high intensity (>20mm h$^{-1}$) and low duration (<20 h) (Fig. 6c and 6e). Accordingly, our CART analysis identifies a higher threshold of rainfall intensity-duration ratio to divide the stable and failed slopes in the urbanised case (Fig. 7b). In landslide analysis, so-called minimum rainfall thresholds are defined as the combinations of rainfall intensity (I) and duration (D) above which we would expect landslides starting to occur. These thresholds are generally expressed by a power law relationship $I = \gamma D^{\alpha}$ (Guzzetti et al. 2007), and they are constructed based on inventories of observed landslides and the rainfall
that triggered them (e.g. Caine, 1980; Larsen and Simon, 1993; Guzzetti et al., 2007). Many countries in the humid tropics have limited empirical data on landslides, and therefore it would be useful to be able to generate such thresholds from stochastic analyses of the type we performed here. To demonstrate how this could be done, we applied a multi-objective optimisation method to our sample of stochastically generated slopes (details about our approach in Section 4 of the supplementary document). We do not use the more commonly employed frequentist methods (Brunetti et al., 2010; Melillo et al., 2018),
because the high frequency of slopes failed for high intensity and high duration events would strongly bias the position of the threshold. Figure 8a and 8b show the calculated thresholds on a log-log scale, respectively for the urbanised and not urbanised case (red lines). In both cases, 99.9% of the failed simulations fall above them. The thresholds present the typical descending trend found in empirical analysis, for which lower rainfall intensities are needed to trigger a landslide when rainfall durations increase. The fact that this trend can be replicated by our synthetic simulations indicates that CHASM+ and our stochastic
modelling framework are giving realistic hydrological and stability responses to the rainfall forcing.

The higher the intensity and/or the duration of the rainfall event, the more slope failures occur in both cases. However, when informal housing is present, more failures are observed for rainfall duration less than 10 hours (short events - Larsen and Simon, 1993). This pushes down the intercept of the rainfall threshold, as reflected in the change in the coefficients of the power law equations (reported in each figure). The slope of the threshold line (i.e. the exponent of the power law) is also
steeper in the urbanised case, implying the presence of more failures for lower rainfall intensities throughout the duration axis. These results are compatible with the increase of small scale landslides previously commented (failure depths less than cut slope's height): to reach saturation at shallow depths, relatively low rainfall intensities and durations can be sufficient to initiate





slope failure. Figure 8c confirms this assumption: when slopes are urbanised (black dots), failures tend to occur with smaller radius of slip surface and for higher values of intensity/duration ratio. The findings reflect the empirical evidence in low income

communities which report a high frequency of small scale landslides, particularly associated with cut slopes, for high intensity and short duration events ('the everyday disasters'- Bull-Kamanga et al., 2003). Finally, we compare our results with the empirical rainfall threshold proposed by Larsen and Simon (1993) for Puerto Rico, which is based on landslide inventories that also include failures observed on slopes modified by construction activities (mainly slope cuts for road network, see Larsen and Parks, 1997). When informal urbanisation is considered, the two thresholds are almost overlapping (Fig. 8b). This

reinforces both the potential of using mechanistic models within a stochastic framework to generate synthetic thresholds in data scarce locations, and the possibility to use the resulting thresholds for regions of the humid tropics with similar geophysical, climatic and urban properties.

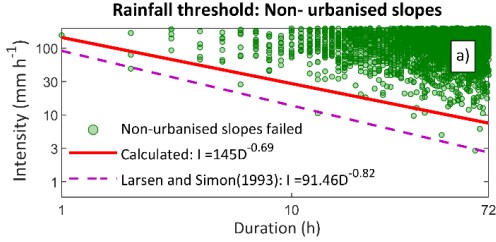

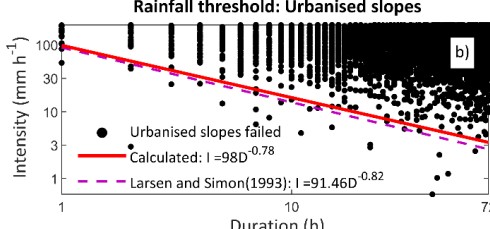

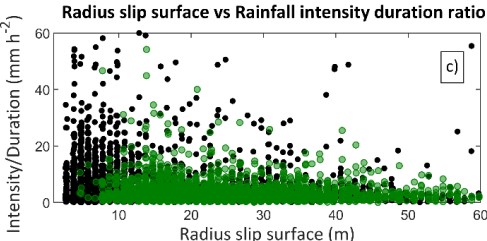

**Figure 8: In figure (a) and (b) the red line represent the minimum rainfall thresholds calculated from our stochastic sample**
**(99.9% of the failed slopes in the sample are above the thresholds). Figure (c) represents the radius of the slip surfaces of the recorded landslides plotted against the corresponding triggering rainfall intensity/duration ratio. Note as in (a) and (b) the x and y axis are in logarithmic base 10 scale, but the notation is linear for an easier readibility.**

**4.3 Guidelines for landslide mitigation actions to tackle the main instability drivers**

The identification of the main instability drivers and their thresholds can contribute to create objective rules to classify slopes
as hazardous in a region with scarce data availability. For example, in Saint Lucia our analysis suggests that slopes with effective cohesion of layer 1 less than 12kPa and thickness less than 2.5m are particularly at risk for rainfall events with intensity/duration ratio greater than 1.06 m $h^{-2}$ (Fig. 7b). These rules can shape look up tables or priority ranking to classify man made slopes as dangerous (Anderson and Lloyd, 1991; Cheng, 2009). Figure 5 shows that only few input factors particularly influence slope stability with or without urbanisation. These are effective cohesion and thickness of the layer 1


(residual soil), slope angle, and rain intensity and duration. The other input factors might have a smaller direct or indirect effect, but they are not dominant. This is an expected finding in global sensitivity analysis (Wagener and Pianosi 2019), despite different outputs (e.g. the timing of the failure) might be sensitive to different input factors (e.g. variations in the moisture suction curves, as demonstrated in the supplementary material S1.2). The identification of these main landslide drivers helps addressing data acquisition efforts, while the comparison between urbanised and not urbanised simulations quantifies the

different relative role (e.g. weight) of preparatory factors in landslide susceptibility assessment when informal urbanisation is present. For example, a weighted average of natural and cut slope angle can be used to identify areas (not) at risk.

  All the results presented are subjected to the assumptions made in our study. The large variation of some of the input factors can lead to overestimating the hazard. Almeida et al. (2017), for example, varied the slope angles between 27 and 30 degrees (instead of between 20 and 45 degrees), and hence found a lower value of the cohesion/thickness ratio to separate stable and

failed slopes than we found (in the non-urbanised case). Data acquisition can help to reduce these uncertainties. However, when data are not available, our approach allows for the identification of so called 'low regrets' mitigation measures, i.e. actions that have a positive impact on slope stability regardless of the uncertain factors. According to our analysis, the most effective action would be avoiding slope cutting, since it resulted as the most detrimental urban construction activity for slope stability. However, this is of scarce utility since informal housing often outstrips urban regulations (Fekade, 2000). Better

hazard awareness and construction practices should be therefore suggested. These include for example reducing surface water infiltration on slopes, especially when the topsoil layers intersect the cut slope and the resulting perched water tables reduce shear strength in a critical location. Slope surface and subsurface drainage can be designed to reduce the infiltration of rainwater to a level that, in effect, reduces the total rainfall intensity below the rainfall threshold calculated. Another cost-effective landslide mitigation strategy can be the planting of deep rooting grasses, shrubs or small trees which increases slope strength

(e.g. slope cohesion) in the top couple of metres of soil and also reduces soil moisture content though root-water uptake and evapotranspiration (Holcombe et al., 2016; Ng et al., 2011; Wilkinson et al., 2002).

  Finally, Fig. 8b shows that when slopes are urbanised, high intensity, short duration rainfall events lead to an increased number of small-scale landslides (failure depths less than 4m, Fig. 6b and radius of slip failure less than 10m, Fig. 8c). Future climate change could potentially increase the frequency of intense precipitation events (e.g. O'Gorman and Schneider, 2009), and

therefore the occurrence of these type of landslides in informal communities. However, if small scale failures produced by anthropogenic factors are neglected in the calculation of rainfall thresholds, also current rainstorms events could be excluded as triggering factors (Crozier, 2010; Mendes et al., 2018). Small scale, high frequency landslide events might not lead to major disasters, but they are increasingly seen as indicators of risk accumulation, detrimental for disaster resilience and economic development (Bull-Kamanga et al., 2003). For this reason, these types of landslides deserve a greater attention from the

scientific community.

### 5. Conclusions

  We include informal housing into slope stability analysis using a newly extended version of the mechanistic model CHASM in a Monte Carlo framework. In this way, we consider uncertainties due to both poorly known slope properties and potential future changes in urban and climate conditions. We demonstrate that informal housing increases landslide hazard and that

slope cutting is the most detrimental construction activity, when compared to vegetation removal, lack of roof gutters and presence of water leaks. The presence of informal housing also modifies the relative role that natural slope angle, soil cohesion and soil thickness have in maintaining slopes stable, with increased hazard occurrence for low values of these three main landslide drivers. CART analysis identifies the thresholds of input factors separating stable and unstable slopes. These thresholds can be used as objective criteria for guiding local engineers in identifying slopes at risk, deducing landslide

mitigation actions, as well as targeting data acquisition to reduce model prediction uncertainty. Moreover, this analysis allows



for the estimation of critical rainfall thresholds at which slope failure is predicted to occur. This rainfall threshold is lower when informal housing is present, with an increased number of small scale landslides (+85%, with failure depth less than 4m and radius of slip surface less than 10m) for high intensity and short duration events. The rainfall threshold resulting from the urbanised slopes is comparable to the one proposed by Larsen and Simon (1993) for the region of Puerto Rico, suggesting its

potential validity also for other similar (data scarce) regions of the humid tropics.

Future work will seek to vary the properties that were kept constant in this study, such as the degree of urbanisation and house dimensions, to evaluate their significance for slope stability. This might confirm the importance of household water management such as roof-guttering, leaking water supply pipes and septic tanks when the number of households is increased. Analysis of slopes where slope cutting is replaced by other possible construction techniques (such as houses suspended on pile

foundations) can identify whether the construction of future hillside settlements could be done in a manner less detrimental to slope stability. Different bioengineering techniques to mitigate hazard likelihood could also be modelled and their effectiveness evaluated. Finally, we seek to transfer the thresholds found in our CART analysis into spatial scale susceptibility maps in order to identify slopes at higher risk within low income urban settlements. This would confirm whether the areas suggested to be most hazardous correspond to areas where more landslides have been observed.


**Supplement**. The supplement related to this article is available online at:

**Author contribution.** EB performed background research, performed the computations and analysis, and wrote the paper. EH, FP and TW supervised the entire study in all stages, discussed the results and contributed to the final manuscript.


**Competing interests.** The authors declare that they have no conflict of interest.

**Acknowledgements**. Existing MATLAB codes from Susana Almeida and Rose Hen Jones have been used for this analysis. We thank Prof. Dave Petley (University of Sheffield) for giving permission to use one of the pictures for Kalimpong, India

(http://savethehills.blogspot.com/). Funding: The first author was supported by an EPSRC PhD studentship. Partial support to TW was provided by a Royal Society Wolfson Research Merit Award [WM170042]. FP is partially funded by the Engineering and Physical Sciences Research Council (EPSRC) "Living with Environmental Uncertainty" Fellowship [EP/R007330/1].

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
