# Peer review of "Including informal housing in slope stability analysis – an application to a data-scarce location in the humid tropics"

_Natural Hazards and Earth System Sciences, 2020_

## Referee Comment (RC1) · Anonymous Referee #1 · 18 Aug 2020

nhess-2020-207

Including informal housing in slope stability analysis–an application to a data-scarce location in the humid tropics

This paper studies the effect of informal housing on the slope stability using an improved mechanistic model CHASM (Combined Hydrology and Stability Model). This is an interesting topic, although this is rarely accounted for landslide hazard assessment. Hence, the reviewer does not suggest the current manuscript for publication. The manuscript needs a major revision.

Some comments for the revision. (1) The study site map should be added to the

reviewed manuscript. Meanwhile, the basic geological setting and rainfall information may be helpful to readers. (2) The thickness of the soil layer is crucial to the model calculation. How to consider the question in the improved modelïij§ (3) According to the reviewer's knowledge, the point water sources from informal housing may be closely related to preferential flow practically. Does your new model take into account the preferential flow?

---

## Referee Comment (RC2) · Anonymous Referee #2 · 31 Aug 2020

**General comment**

The manuscript (MS) deals with modeling possible impacts of informal urbanization on the hydrologic and geo-mechanical response of hillslopes, also with the aim at understanding which of the factors of such an urbanization process are the most detrimental for slope stability. The modeling is built as an extension of a previously released model (CHASM).

I really enjoyed reading the MS, which is well written and structured. The supplementary material explains in detail the CHASM+ model and other aspects of the MS, and it is really an added value to the main text.

[Figure]

From a general standpoint, the conclusion that slope cutting is the most detrimental among the other factors included in the modeling could be somewhat expected/or reached without the use of the massive modeling in the paper. However, I think that the main contribution given by this MS is that the model enables to QUANTIFY the response of the hillslope to the most important factors of informal urbanization and that it presents the application of some interesting statistical techniques to resume and communicate the main results of the modeling.

Processes are represented in a somewhat simplified manner, but still the resulting model is quite complex and has several input parameters. Perhaps one could argue about some of the choices made in the model and the definition of the parameters' probability distributions (see also referee 1), but my opinion is that the authors have made all those choices in the most reasonable manner possible.

For all the reasons above, I finally think this is a very good work, and my opinion is that that the MS can be accepted after minor revisions. In the following I provide just some suggestions to improve it.

**Specific comments**

L 83 The MS "promises" that somehow the modeling exercise will take into account climate change. I think this is quite weak in the analysis presented. The authors should discuss a little if climate change projections could be used to define future values of rainfall based on Representative concentration scenarios and simulations by Regional/Global climate models, and mention literature on the subject: e.g. https://doi.org/10.1016/j.jhydrol.2016.02.007, https://doi.org/10.1016/J.JHYDROL.2018.10.036

LL 198-200 The water table height is varied between 0 and 90 % of the slope height. This seems a quite wide range. Perhaps the reasons for this choice could be better explained.

[Figure]

L 234 Perhaps a reference explaining the Latin Hypercube sampling technique can be useful for readers.

Section 4.2 and LL 263-275 of the supplement: The objectives of the multi-optimization are quite unusual. Perhaps in this case, an optimization based on ROC (receiver operating characteristics) analysis (i.e.: True and false positives/negatives) could have been employed and would have been more meaningful. At least, literature in the subject should be mentioned: e.g. https://doi.org/10.1007/s10346-020-01420-8, https://doi.org/10.1029/2012JF002367, https://doi.org/10.5194/hess-18-4913-2014

Fig S1 (supplement): Panel (a) is repeated in panel (b), so perhaps it could be removed. Possibly add to the plot the rainfall time series (cumulated sum)

Section S1. Perhaps the case of houses WITH gutters should be explained.

**Technical corrections**

L60 of the supplement: CHAMS -> CHASM (check the entire MS)

L137 the comma before "ranges" seems not necessary

---

## Author Comment (AC1) · 11 Sep 2020

We thank Reviewer 1 for taking the time to read our paper. We think the Reviewers' comments can be addressed in a revised manuscript as follow.

Comment (1): The study site map should be added to the C1 NHESSD Interactive comment Printer-friendly version Discussion paper reviewed manuscript. Meanwhile, the basic geological setting and rainfall information may be helpful to readers.

The methodology applied allows evaluation of the probability of failure of slopes for which there is scarce and/or uncertain data. Rather than referring to a specific site with

measured geometry, urbanisation, soil and rainfall data, we stochastically generate tens-of-thousands of possible slope cross-sections that represent the population of slopes that might be observed in the case study region (using data from literature and previous fieldwork). Adding a map could therefore be misleading. Though, we will modify the text of the introduction to render the concept of stochastic generation of slopes clearer to the reader. Additionally, we will replace the term "site" with "case study" or "case study region" to avoid any confusion. Furthermore, we will add, as suggested, more information on the type of climate (humid tropical) and on the type of soil and weathering grade usually found in the region in section 2.1, where other information about the geological setting are given.

Comment (2): The thickness of the soil layer is crucial to the model calculation. How to consider the question in the improved model

We agree with the reviewer on the importance of the thickness of the soil layers. Indeed, our results confirm this point via both the sensitivity analysis and CART. Soil thickness is considered in our modelling as an uncertain input factor and it is stochastically varied within a reasonable range, deduced from previous fieldwork. We are thus not completely sure about what the Reviewer means by the statement "how to consider the question [which question?] in the improved model.

Comment (3): According to the reviewer's knowledge, the point water sources from informal housing may be closely related to preferential flow practically. Does your new model take into account the preferential flow?

We did not include preferential flows in CHASM+. We agree with the Reviewer that leaking pipes and buried tanks can induce soil pipe erosion in response to increasing water inputs. This could be simulated for example with a dual permeability model, but then it would be difficult to implement the pore pressure calculated into the slope stability model. Furthermore, the inclusion of preferential flows requires the definition of additional input factors which may be difficult in data-scarce contexts. So, given

the spatial scale, the purpose of the analysis and the data available, we believe that the current CHASM+ representation is sufficient to depict landslide initiation due to flow accumulation around the point water source. We will add this discussion in the supplement (S1.1). We will also add in the same section a comment about the fact that a dynamic change in the hydraulic properties due to the water leaked is anyway already taken into account by the model, given the way that CHASM represent hydrological processes.
* * *

---

## Author Comment (AC2) · 11 Sep 2020

We thank Reviewer 2 for taking the time to read our paper, for the positive comments and for recognising the contribution this research makes. We think we can revise our manuscript and address the specific comments as follow.

Comment (1): L 83 The MS "promises" that somehow the modelling exercise will take into account climate change. I think this is quite weak in the analysis presented. The authors should discuss a little if climate change projections could be used to define future values of rainfall based on Representative concentration scenarios and simulations by Regional/Global climate models, and mention literature on the subject.

[Figure]

We understand where the Reviewer comment is coming from, as we have adopted a perhaps less common approach to account for climate change in our modelling. In the approach we adopt, sometimes referred to as 'bottom-up' (Groves and Lempert, 2007; Wilby and Dessai, 2010) we do not choose a single climate projection scenarios to define future values of rainfall and propagate them through the modelling chain ('top-down' approach) but rather we uniformly increase the severity of observed rainfall events and use CART to find those combinations of rainfall (and other uncertain input factors) that would produce unwanted outcomes (slope failure in our case). We therefore explore the feasible rainfall space widely, rather than focusing on the potentially more likely space covered (in terms of rainfall intensity and duration) of one or more scenarios (even though we will include this scenario space). In this way we can 1) quantify the effects of other uncertainties (such as uncertain soil properties) compared to climate uncertainty; 2) identify for which values of rainfall intensity and duration landslide hazard starts to significantly increase. These threshold values may then be compared to GCMs projections for a specific place, in order to assess the chances that they be exceeded in the future. We recognise the point is not clear in the manuscript and it requires a better explanation. We will modify part of the introduction to include the above discussion and references.

Comment (2): LL 198-200 The water table height is varied between 0 and 90 % of the slope height. This seems a quite wide range. Perhaps the reasons for this choice could be better explained.

The wide range aims to represent the variability across the ensemble of slopes that can be found in our study region. We then use CART to define thresholds of water table height above which slope failure is more likely to occur. These threshold values can be then compared with levels of water table height of a particular slope and in a particular moment to assess its landslide probability. We will better specify the stochastic generation of the water table in section 2.4

Comment (3): L 234 Perhaps a reference explaining the Latin Hypercube sampling

technique can be useful for readers.

Agreed. We will add a reference explaining the Latin Hypercube sampling

Comment (4) : Section 4.2 and LL 263-275 of the supplement: The objectives of the multi-optimization are quite unusual. Perhaps in this case, an optimization based on ROC (receiver operating characteristics) analysis (i.e.: True and false positives/negatives) could have been employed and would have been more meaningful. At least, literature in the subject should be mentioned.

Optimization based on ROC analysis could have been an option, though we think our approach is also suitable given that our aim was to essentially identify the two parameters of the minimum rainfall threshold line. We already cite a study that employs ROC analysis and a review where it is mentioned (Staley et al. 2013 and Segoni et al. 2018), but a more explicit reference will be made in section 4 of the supplement.

Comment (5): Fig S1 (supplement): Panel (a) is repeated in panel (b), so perhaps it could be removed. Possibly add to the plot the rainfall time series (cumulated sum).

We will modify Figure S1 as suggested

Comment (6): Section S1. Perhaps the case of houses WITH gutters should be explained.

We will explain also the case WITH gutters and not just without in Section S1 as suggested.

Technical corrections L60 (supplement) and L137

Thank you for identifying these. We will address these typographic and grammatical errors and check the whole manuscript and supplement.

---

## Author Response (AR1)

**Response to Reviewer #1**

We thank Reviewer 1 for taking the time to read our paper. We think the Reviewers' comments can be addressed in a revised manuscript as follow. The comments made by the Reviewer are in black, and our responses are in blue. The italics highlights the additions made to the revised manuscript. Line numbers refer to the revised manuscript.

5   This paper studies the effect of informal housing on the slope stability using an improved mechanistic model CHASM (Combined Hydrology and Stability Model). This is an interesting topic, although this is rarely accounted for landslide hazard assessment. Hence, the reviewer does not suggest the current manuscript for publication. The manuscript needs a major revision.

Some comments for the revision.

10   (1)   The study site map should be added to the C1 NHESSD Interactive comment Printer-friendly version Discussion paper reviewed manuscript.

Authors' reply: The methodology applied allows evaluation of the probability of failure of slopes for which there is scarce and/or uncertain data. Rather than referring to a specific site with measured geometry, urbanisation, soil and rainfall data, we stochastically generate tens-of-thousands of possible slope cross-sections that represent the population of slopes that might be

15   observed in the case study region (using data from literature and previous fieldwork). Adding a map could therefore be misleading. However, to render the concept of stochastic generation of slopes clearer to the reader we modified the text of the introduction as follows (L.83):

*"A sample of tens-of-thousands of rainfall events and slopes was stochastically generated from these distributions and simulated in CHASM."*

Additionally, we replaced the term "site" with "case study" to avoid any confusion. Furthermore, we added, as suggested, more information on the type of climate (humid tropical) and on the type of soil and weathering grade usually found in the region in section 2.1, where other information about the geological setting are given.

25   We replaced the title of section 2.1 (L.137) with:

*"Case study: Saint Lucia, Eastern Caribbean"*

We added more explicit information on the climate and geological settings, in section 2.1 (L.138):

*"Saint Lucia is an Eastern Caribbean island with a humid tropical climate. The main landslide trigger is rainfall, and shallow*

30   *rotational landslides dominate on both steep and shallow slopes (Van Westen, 2016; Anderson and Holcombe, 2013). The geology is almost entirely comprised of volcanic bedrock and deep volcanic deposits. Due to the tropical climate, these volcanic parent materials are subjected to deep weathering, which decreases their strength and increases landslide susceptibility. The strata of a typical slope cross section comprise weathered residual soils overlying decomposed rock and volcanic bedrock. These three types of strata typically correspond respectively to the weathering Grade V-VI, Grade II-IV and*

35   *Grade I-II, of the Hong Kong Geotechnical Engineering Office weathering grade classification (GEO, 1988). There is a high variability in terms of engineering soils, but they can broadly classify as fine grained soils such as silty clays, clayey silts and sandy clays (DeGraff, 1985). The combination of tropical climate, steep topography and volcanic geology render the region particularly susceptible to rainfall-triggered landslides. Furthermore, landslide risk is increased by informal housing which occupy steep slopes and employ unregulated engineering practices (WB/GFDRR 2012, p. 226-235)".*

40   *Reference: DeGraff, J. F.: Landslide hazard on St. Lucia, West Indies- Final Report: Washington D. C., Organization of American States, 1985.*

(2) The thickness of the soil layer is crucial to the model calculation. How to consider the question in the improved modeling

45 Authors' reply: We agree with the reviewer on the importance of the thickness of the soil layers. Indeed, our results confirm this point via both the sensitivity analysis and CART. Soil thickness is considered in our modelling as an uncertain input factor and it is stochastically varied within a reasonable range, deduced from previous fieldwork. We are thus not completely sure about what the Reviewer means. No modifications were made in the main manuscript to address this comment.

50 (3) According to the reviewer's knowledge, the point water sources from informal housing may be closely related to preferential flow practically. Does your new model take into account the preferential flow?

Authors' reply: We did not include preferential flows in CHASM+. We agree with the Reviewer that leaking pipes and buried tanks can induce soil pipe erosion in response to increasing water inputs. We suggest that the wide range of hydraulic conductivity values sampled in the stochastic modelling approach could be assumed to account for the potential effects of

55 increased hydraulic conductivity at the grid-cell resolution (i.e as a lumped or effective permeability that averages the $K_{sat}$ for the preferential flow paths and the $K_{sat}$ of the surrounding soil in each cell). Based on our benchmarking study of the new CHASM+ point-water source functionality (see Supplementary material) we believe that the current CHASM+ representation is sufficient to depict landslide initiation due to flow accumulation from the point water source. We added this discussion in the supplement (S1.1, L.39) as follow:

60

*"Leaking pipes and buried tanks can induce soil pipe erosion in response to increasing water inputs. This could be simulated for example with a dual permeability model, but then it would be difficult to implement the pore pressure calculated into the slope stability model (Bogaard and Greco, 2016). Furthermore, the inclusion of preferential flows requires the definition of additional input factors which may be difficult in data-scarce contexts. So, given the spatial scale, the purpose of the analysis*

65 *and the data available, the current CHASM+ representation can be considered sufficient to depict landslide initiation due to flow accumulation around the point water source".*

*Reference: "Bogaard, T. A. and Greco, R.: Landslide hydrology: from hydrology to pore pressure, Wiley Interdiscip. Rev. Water, 3(3), 439–459, doi:10.1002/wat2.1126, 2016."*

70

We also added in the same section (L.15) a comment about the fact that a dynamic change in the hydraulic properties due to the water leaked is anyway already taken into account by the model, given the way that CHASM represents hydrological processes.

*"When water is added into the cell, the moisture content increases. The unsaturated hydraulic conductivity, which depends on*

75 *the moisture content, also increases and is iteratively calculated with the Millington-Quirk formulation (Millington and Quirk, 1959). The maximum value is reached when soil is saturated (saturated hydraulic conductivity, fixed by the user).*

80

85

**Response to Reviewer #2**

We thank Reviewer 2 for taking the time to read our paper and helpful suggestions to clarify the manuscript. The comments made by the Reviewer are in black, and our responses are in blue. The italics highlights the additions made to the reviewed manuscript. Line numbers refer to the revised manuscript.

90 General comment:

The manuscript (MS) deals with modeling possible impacts of informal urbanization on the hydrologic and geo-mechanical response of hillslopes, also with the aim at understanding which of the factors of such an urbanization process are the most detrimental for slope stability. The modeling is built as an extension of a previously released model (CHASM). I really enjoyed reading the MS, which is well written and structured. The supplementary material explains in detail the CHASM+ model and

95 other aspects of the MS, and it is really an added value to the main text.

This paper studies the effect of informal housing on the slope stability using an improved mechanistic model CHASM (Combined Hydrology and Stability Model). This is an interesting topic, although this is rarely accounted for landslide hazard assessment. Hence, the reviewer does not suggest the current manuscript for publication. The manuscript needs a major revision.

100 From a general standpoint, the conclusion that slope cutting is the most detrimental among the other factors included in the modeling could be somewhat expected/or reached without the use of the massive modeling in the paper. However, I think that the main contribution given by this MS is that the model enables to QUANTIFY the response of the hillslope to the most important factors of informal urbanization and that it presents the application of some interesting statistical techniques to resume and communicate the main results of the modeling. Processes are represented in a somewhat simplified manner, but

105 still the resulting model is quite complex and has several input parameters. Perhaps one could argue about some of the choices made in the model and the definition of the parameters' probability distributions (see also referee 1), but my opinion is that the authors have made all those choices in the most reasonable manner possible. For all the reasons above, I finally think this is a very good work, and my opinion is that that the MS can be accepted after minor revisions. In the following I provide just some suggestions to improve it.

110 Specific comments:

(1) L 83 The MS "promises" that somehow the modeling exercise will take into account climate change. I think this is quite weak in the analysis presented. The authors should discuss a little if climate change projections could be used to define future values of rainfall based on Representative concentration scenarios and simulations by Regional/Global climate models, and

115 mention literature on the subject: e.g. https://doi.org/10.1016/j.jhydrol.2016.02.007, https://doi.org/10.1016/J.JHYDROL.2018.10.036

Authors' reply: We understand where the Reviewer is coming from with this comment as we have adopted a perhaps less common approach to account for climate change in our modelling. In the approach we adopt, sometimes referred to as 'bottom-

120 up' (Groves and Lempert, 2007; Wilby and Dessai, 2010), we do not choose a single climate projection scenarios to define future values of rainfall and propagate them through the modelling chain ('top-down' approach) but rather we uniformly increase the severity of observed rainfall events and use CART to find those combinations of rainfall (and other uncertain input factors) that would produce unwanted outcomes (slope failure in our case). We therefore explore the feasible rainfall space widely, rather than focusing on the potentially more likely space covered (in terms of rainfall intensity and duration) of one or

125 more scenarios (even though we will include this scenario space). In this way we can 1) quantify the effects of other uncertainties (such as uncertain soil properties) compared to climate uncertainty; 2) identify for which values of rainfall intensity and duration landslide hazard starts to significantly increase. These threshold values may then be compared to GCMs projections for a specific place, in order to assess the chances that they could be exceeded in the future. We recognise the point

is not clear in the manuscript and it requires a better explanation. We will include in the introduction the above discussion and references as follow (L.84):

*"By this approach the possible effects of climate change were explored widely, instead of focusing on one (or a few) climate projection scenarios (such as those provided by downscaled generalised circulation models) propagated through the modelling chain (Groves and Lempert, 2007; Wilby and Dessai, 2010). This strategy can be extended to include the exploration of both feasible climate as well as feasible land use futures (Singh et al., 2014)".*

*Reference: Groves, D. G. and Lempert, R. J.: A new analytic method for find- ing policy-relevant scenarios, Global Environ. Chang., 17, 73– 85, doi:10.1016/j.gloenvcha.2006.11.006, 2007.*

*Wilby, R. L. and Dessai, S.: Robust adaptation to climate change, Weather, 65, 180–185, doi:10.1002/Wea.543, 2010.*

*Singh, R., Wagener, T. Crane, R. Mann, M. E. and Ning L.: A vulnerability driven approach to identify adverse climate and 715 land use change combinations for critical hydrologic indicator thresholds: Application to a watershed in Pennsylvania, USA, Water Resour. Res., 50, 3409–3427, doi:10.1002/2013WR014988, 2014*

We also added a figure (Fig.2 and related text) representing the feasibility space used for the stochastic generation of the rainfall events. The figure should clarify how future climate events (i.e. extreme combinations of rainfall intensity and duration) are considered in the analysis (L. 207)

*"The model is forced with rainfall events which are specified in terms of their duration (in hours) and hourly intensity. The aim is to create both rainfall events that have been observed in the past, and rainfall events that might occur in the future (e.g. with higher intensity and duration than observed historically). To constrain the rainfall variability space, we use the intensity-duration-frequency (IDF) relationships derived from a Gumbel analysis of 40-years of daily rainfall data from weather stations across the island by Klohn-Crippen (1995) (Fig.2). From these IDFs we derive a range of rainfall intensity between 0 and 200 mm h-1, and a range of rainfall duration between 0 and 72 h. We then sample independently from the two uniform distributions, thus obtaining combinations of intensity and duration that might have been observed in the past (light grey area in Fig. 2) or not (dark grey area in Fig. 2)."*

(2) LL 198-200 The water table height is varied between 0 and 90 % of the slope height. This seems a quite wide range. Perhaps the reasons for this choice could be better explained.

Authors' reply: The wide range aims to represent the variability across the ensemble of slopes that can be found in our study region. We then use CART to define thresholds of water table height above which slope failure is more likely to occur. These threshold values can be then compared with levels of water table height of a particular slope and in a particular moment to assess its landslide probability. We better specify the stochastic generation of the water table in section 2.4, L. 204:

*"This water table height is varied between 0% and 90% of the slope height (H in Fig. 3), to account for its variability across the region and for the variability of the initial soil moisture conditions due to antecedent rainfall events."*

(3) L 234 Perhaps a reference explaining the Latin Hypercube sampling C2 NHESSD Interactive comment Printer-friendly version Discussion paper technique can be useful for readers.

Authors' reply: L.242 *"(McKay et al., 1979)"*

*Reference: "McKay, M. D., Beckman, R. J. and Conover, W. J.: Comparison of three methods for selecting values of input variables in the analysis of output from a computer code, Technometrics, 21(2), 239–245, doi:10.1080/00401706.1979.10489755, 1979".*

(4) Section 4.2 and LL 263-275 of the supplement: The objectives of the multi-optimization are quite unusual. Perhaps in this case, an optimization based on ROC (receiver operating characteristics) analysis (i.e.: True and false positives/negatives) could have been employed and would have been more meaningful. At least, literature in the subject should be mentioned: e.g. https://doi.org/10.1007/s10346-020-01420-8, https://doi.org/10.1029/2012JF002367, https://doi.org/10.5194/hess-18-4913-2014

Authors' reply: Optimization based on ROC analysis could have been an option, though we think our approach is also suitable given that our aim was to essentially identify the two parameters of the minimum rainfall threshold line. We already cite a study that employs ROC analysis and a review where it is mentioned (Staley et al. 2013 and Segoni et al. 2018). We make a more explicit reference in L. 239 of the supplement:

*"An alternative to this approach could be to use a (single-objective) optimization based on ROC (receiver operating characteristics), where false positives and negatives (represented in this case by the simulated landslides below the threshold and simulated stable slopes above the threshold) are minimised (Gariano et al., 2015; Staley et al., 2013)".*

*Reference: Gariano, S. L., Brunetti, M. T., Iovine, G., Melillo, M., Peruccacci, S., Terranova, O., Vennari, C. and Guzzetti, F.: Calibration and validation of rainfall thresholds for shallow landslide forecasting in Sicily, southern Italy, Geomorphology, 228, 653–665, doi:10.1016/j.geomorph.2014.10.019, 2015.*

(5): Fig S1 (supplement): Panel (a) is repeated in panel (b), so perhaps it could be removed. Possibly add to the plot the rainfall time series (cumulated sum).

Authors' reply: We have modified Figure S1 as suggested

(6): Section S1. Perhaps the case of houses WITH gutters should be explained.

Authors' reply: We explained also the case WITH gutters in L37 of section S1.1:

*"If gutters are present the rainwater intercepted by the roof is deleted, consequently decreasing the rainfall rate infiltrating into the slope".*

Technical corrections L60 (supplement) and L137

Authors' reply: We have addressed these typographic and grammatical errors and check the whole manuscript and supplement.

Track changes version of the main manuscript: please note that there might be some minor differences between the .pdf uploaded and the text below (some of the changes have been accepted by mistake but they do not regard the reviewers' comments).

[revised manuscript text omitted]
 GCMs) to propagated through the modelling chain ('top-down' approach), but rather by uniformly increasing the severity of the rainfall event, i.e. increasing the rainfall intensity and duration ('bottom-up' approach (Groves and Lempert, 2007; Wilby and Dessai, 2010). This strategy can be extended to include the exploration of both feasible climate as well as feasible land use futures (Singh et al., 2014). ). The rainfall events and a population of tens-of-thousands slopes that might be observed in the region in the present or in the future were stochastically generated and simulated in CHASM. 
[revised manuscript text omitted]
 rainfall intensityies and duration than historically observeds historically). To constrain the rainfall feasibilityvariability space,are based on we use the intensity-duration-frequency (IDF) relationships derived from a Gumbel analysis of 40-years of daily rainfall data from weather stations across the island by Klohn-Crippen (1995) (Fig.2a). From these IDFs we derivfine a ranges of rainfall intensity from between 0 to and 200 mm h$^{-1}$, and a range of rainfall duration from between 0 to and 72 h. which areWe then sampled independently from the two uniform distributions. In this way, thus obtaining combinations of rainfall intensity and duration both observed (light grey area in Fig. 2b) and not observed in the past (dark grey area in Fig. 2b) in the past are stochastically generated. 
[revised manuscript text omitted]

975

980

985

990

995

1000

1005

1010

1015 Supplement reviewed. Highlighted the changes related to the reviewers comments. Please refer to the .pdf uploaded as supplement for the fully reviewed version.

**S1 Slope water management in the Combined Hydrology And Stability Model, CHASM**

Section 1 is divided in two parts: the first describes the new functionality developed in CHASM representing slope water management (S1.1); the second part illustrates its benchmark against another slope stability software (S1.2).

1020 **S1.1 Description of urban slope water management in CHASM+**

We have developed new functionality in the CHASM code (which we are calling 'CHASM+') which is now able to simulate three additional processes: water leaks from buried septic tanks, leaks from superficial pipes, and the effect of houses without roof gutters discharging rainwater from their roofs onto the slope.

**Leaking Septic tanks:** the user can determine the position of the tanks, their dimensions (width and depth), the leakage rate

1025 ($m^3 s^{-1}$) and the type of leakage (local or evenly distributed). Considering that the slope cross section is represented with a mesh of columns and cells, a tank will occupy some of these cells according to its dimensions and position. These cells are modelled as being impermeable and heavier than the surrounding soil. The water leakage is added to the moisture content of the cells underneath the tank, through the following Water Balance equation (S1):

$$\frac{\partial \theta}{\partial t} = \frac{\partial (Q + Q_{leak})}{\partial z} \tag{S1}$$

1030 where, $\theta$ is the moisture content, changing over time according to the water flow $Q$, and $Q_{leak}$ represents the water leaked by the tank, which is constant throughout the simulation time. When water is added into the cell, the moisture content increases. The unsaturated hydraulic conductivity, which depends on the moisture content, also increases and is iteratively calculated with the Millington-Quirk formulation (Millington and Quirk, 1959). The maximum value is reached when soil is saturated (saturated hydraulic conductivity, fixed by the user).

1035 Note, buried leaking pipes can also be simulated by using this option by not considering the load of the tank.

**Leaking Pipe on the slope surface:** we want to simulate pipes discharging water onto the slope surface. This can be due by low pipe maintenance or when water collectors are poorly designed and not properly connected to formal drainage or sewerage system (Ortuste, 2012). In CHASM the slope cross section is represented by a two-dimensional mesh of columns and cells for the purposes of the hydrology calculations. Therefore, the water leaked by the pipe is added to the surface water of the column

1040 of the slope where the pipe is positioned. This water infiltrates into the slope according to the infiltration capacity of the top cell of that column, which is a function of its hydraulic conductivity. The water that does not infiltrate because exceeds the infiltration capacity, is stored on the surface as ponding water. The maximum storage of ponding water is determined by the user as detention capacity of that cell. If the ponding water exceeds this value, it is removed from the calculation because surface water runoff is not included in the CHASM hydrology scheme. The leakage when present is constant throughout the

1045 simulation time.

**Houses without gutters:** Where houses are present, rainfall does not reach the top cell of the slope underneath the house, and the amount of rain intercepted by the roof is calculated and discharged onto (added to) the top cells of the slope to the sides of the house. If the roof is dual pitch, half of the intercepted rain is discharged upslope and half downslope of the house, and it is equal to the rainfall rate multiplied by half of the roof area. This means that the surface water being added to the cells

1050 immediately adjacent to the house is the sum of the rainfall that would fall in that cell plus the intercepted rainfall discharged from the roof. The same calculation is used for the mono pitch roof, but in this case the rainwater that falls on the roof is entirely discharged downslope or upslope of the house and it is equal to the rainfall rate multiplied by the whole roof area. The

surface water will then infiltrate into the slope as described for the case of the leaking pipe. If gutters are present, the rainwater intercepted by the roof is deleted, consequently decreasing the rainfall rate infiltrating into the slope.

1055 Leaking pipes and buried tanks can induce soil pipe erosion in response to increasing water inputs. This could be simulated for example with a dual permeability model, but then it would be difficult to implement the pore pressure calculated into the slope stability model (Bogaard and Greco, 2016). Furthermore, the inclusion of preferential flows requires the definition of additional input factors which may be difficult in data-scarce contexts. So, given the spatial scale, the purpose of the analysis and the data available, the current CHASM+ representation can be considered sufficient to depict landslide initiation due to

1060 flow accumulation around the point water source.

[revised manuscript text omitted]

- a multi-objective optimisation algorithm for the estimation of the two parameters γ and α of Eq. S3.

The multi-objective optimisation involves minimising or maximising multiple objective functions subject to a set of constrains. In this case, we want to draw a threshold line in the form of Eq. S3 which identifies the space where landslides are recorded. This translates into choosing parameters γ and α of Eq. S3 that satisfy the following two contrasting objectives:

1) maximise the number of (simulated) failed slopes falling above the threshold line (Fig. S9a)
2) minimise the area above the threshold line (Fig. S9b)

To constrain the search to realistic values of rainfall intensity and duration, the optimisation only explores values of γ and α within upper and lower boundaries specified as:

$$\gamma\ [-0.5; -2] \tag{S4a}$$

$$\alpha\ [0.05; 2] \tag{S4b}$$

The range of α so defined includes typical slope values of empirical rainfall thresholds (Guzzetti et al., 2007), while the range of γ is designed to include all the rainfall intensities simulated. To perform the multi-objective optimisation, we used the generic algorithm implemented in the "gamultiobj" function of the Matlab Optimisation Toolbox (R2018a). As any multi-objective optimiser, it produces a set of Pareto-optimal solutions that realise different optimal trade-offs of the two objectives. In this case, 13 possible optimal combinations of (γ, α) are obtained, and among them we (subjectively) chose the one that gives a threshold line with 99.9% of failed simulations above it or, in other words, with 0.1% landslide probability below it. This is the threshold line reported in Fig. 98a,b of the main manuscript. A different choice could be made to determine the threshold line for any exceedance probability level. An alternative to this approach could be to use a (single-objective) optimization based on ROC (receiver operating characteristics), where false positives and negatives (represented in this case by the simulated landslides below the threshold and simulated stable slopes above the threshold) are minimised (Gariano et al., 2015; Staley et al., 2013).

[Figure]

[Figure]

**Figure S9: Illustration of the two objectives functions used in the optimisation, for a given threshold line: (a) maximise the number of failed slopes above the threshold and (b) minimise the area above the threshold.**